# Sex-Heterogeneity on the Association between Dietary Patterns at 4 Years of Age with Adiposity and Cardiometabolic Risk Factors at 10 Years of Age

**DOI:** 10.3390/nu14030540

**Published:** 2022-01-26

**Authors:** Catarina Durão, Milton Severo, Andreia Oliveira, Carla Lopes

**Affiliations:** 1EPIUnit-Institute of Public Health, University of Porto, 4050-600 Porto, Portugal; catarina.durao@nms.unl.pt (C.D.); milton@med.up.pt (M.S.); acmatos@med.up.pt (A.O.); 2Laboratory for Integrative and Translational Research in Population Health (ITR), 4050-600 Porto, Portugal; 3NOVA Medical School, NOVA University of Lisbon, 1169-056 Lisbon, Portugal; 4Public Health and Forensic Sciences and Medical Education Department, University of Porto Medical School, University of Porto, 4200-319 Porto, Portugal

**Keywords:** dietary pattern, cohort, obesity, cardiometabolic risk factors, sex differences

## Abstract

To evaluate the association of dietary patterns (DP) at 4 years with adiposity and cardiometabolic risk factors at 10 years, considering sex-heterogeneity. This prospective analysis included 3823 children enrolled in the population-based birth cohort, Generation XXI (Porto-Portugal, 2005–2006). Diet at 4 years was assessed by FFQ, with three DP being identified: high in energy-dense foods (EDF), intermediate in snacks (snacking), and healthier (reference). BMI at 10 years was considered as the Z-score according to the WHO. Other adiposity indicators—fat mass percentage (FM%), fat mass index (FMI), and waist-to-height ratio (WHtR)—were converted to z-scores using the sample’s sex-specific means and standard deviations, as were the cardiometabolic risk factors (systolic and diastolic blood pressure, lipid profile, and insulin resistance). The associations of DP at 4 years with later adiposity or cardiometabolic factors were estimated by linear regression or by multinomial logistic regression models. In fully adjusted models, the EDF DP was significantly positively associated with the BMI (EDF vs. healthier: β = 0.139; 95% CI: 0.031, 0.246, P-interaction = 0.042) and obesity (OR = 2.68; 95% CI 1.55, 4.63, P-interaction = 0.005) only in girls, among whom, it increased insulin (β = 0.165; 95% CI: 0.020, 0.311) and HOMA-IR (β = 0.159; 95% CI: 0.013, 0.306) at 10 years. An EDF DP at 4 years is associated with later adiposity, insulin, and HOMA-IR in girls.

## 1. Introduction

The prevalence of obesity in children, despite reports of plateauing in some high-income countries, remains high, and is a crucial public health problem, as reflected by global numbers, with 50 million girls and 74 million boys suffering from obesity in 2016 [1]. In Portugal, the prevalence of overweight or obesity among children under 10 years of age was 25% in 2015–2016 [2]. According to the WHO European Childhood Obesity Surveillance Initiative (COSI), data from the fourth round (2015–2017) found a prevalence of overweight of 32% in girls and 29% in boys at the ages of 6–9 years, with southern European countries showing a higher prevalence, although a significant decrease in the prevalence of both overweight and obesity was observed in the last 10 years in Greece, Italy, Slovenia, and Portugal [3].

Obesity is particularly relevant to cardiometabolic health. Cardiovascular disease typically manifests during adulthood, but cardiovascular risk factors may start to emerge during childhood. Cardiometabolic risk factors commonly found in children with overweight or obesity are hypertension, dyslipidemia, increased fasting insulin, and insulin resistance [4]. Also, persistence of overweight or obesity into adulthood is linked to a high cardiovascular risk [5]. As such, considering that both overweight [6] and other cardiometabolic risk factors [7] show tracking into adulthood, prevention at early stages is crucial.

Diet is a key factor for both obesity and other cardiometabolic risk factors [8] and shows tracking into later ages [9]. In addition, an obesogenic environment that increases the consumption of energy-dense micronutrient-poor foods and beverages (EDF) is of particular importance for both obesity [10] and cardiometabolic health during childhood. The dietary intake during preschool age is especially relevant because eating behaviors and food preferences, which influence future diet, start to be established at early stages [11]. However, studies on dietary patterns as early exposures of cardiometabolic health are still scarce [7,8,12].

Sex differences having an influence on adiposity and cardiometabolic health have been previously explored [13,14,15], but a possible sex-heterogeneity on the association of diet with adiposity and cardiometabolic health needs further clarification. We have previously observed sex-heterogeneity on the association of a dietary pattern high in EDF, followed at the age of 4 years with adiposity at 7 years [16], as well as on the association of the protein and glycemic load of preschool children aged 4 years with adiposity and fasting insulin [17] 3 years later. In the current study, we hypothesized that sex-heterogeneity could be observed on the association of dietary patterns practiced during preschool age with later adiposity and cardiometabolic risk factors. As such, the present study aims to evaluate the association of dietary patterns at 4 years of age with adiposity and other cardiometabolic risk factors at 10 years of age, considering differences according to child’s sex.

## 2. Materials and Methods

### 2.1. Participants

This prospective study was based on the population-based birth cohort Generation XXI, assembled between April 2005 and August 2006, that is regularly followed [18]. At baseline, 8647 live newborns and their mothers were recruited at the public maternity units that cover the six municipalities of the metropolitan area of Porto (Northern Portugal), representing 92% of the deliveries of the catchment population. As described elsewhere [16], at child’s age of 4 years, the cohort was invited for evaluation, with 7458 children being reevaluated (86% participation proportion). At 7 years, another evaluation wave took place, reaching 80% participation proportion. Three years later, when children reached their 10th birthday, the cohort was again invited for evaluation (6889 children, 76% participation proportion). All evaluation waves were performed in person, during interviews conducted by trained interviewers.

The current analysis includes information collected prospectively on dietary intake and anthropometrics during three evaluation waves (at 4, 7, and 10 years of age) and considers singleton children with complete Food Frequency Questionnaire (FFQ) and anthropometrics at 4 years of age (evaluated in 2009 and 2010), who are also evaluated at the ages of 7 and 10 years, with thorough data on anthropometrics and body composition (*n* = 4352). Children with conditions that could affect dietary intake (e.g., cerebral palsy or food allergy, *n* = 9), as well as 6 children with celiac disease, were not included. Furthermore, children with incomplete information on both practice of physical exercise and daily screen time at the age of 4 years (*n* = 107), children with incomplete information on maternal BMI at the 4 years wave (*n* = 235), and those with no information on blood pressure at 10 years of age (*n* = 2) were excluded. A total of 3823 participants remained for analyses. At the 10-year evaluation, for a subsample of the cohort, blood samples were collected as described elsewhere [12], with 2811 children having complete information on cardiometabolic risk factors at 10 years of age, such as systolic (SBP) and diastolic blood pressure (DBP), triacylglycerols (TG), high-density lipoprotein cholesterol (HDL-c), low-density lipoprotein cholesterol (LDL-c), glucose, and fasting serum insulin).

The study was conducted according to the guidelines of the Declaration of Helsinki and was approved by the University of Porto Medical School/S. João Hospital Ethics Committee (27 April 2005), as well as by the Portuguese Data Protection Authority (Protocol code 5833, approved on 30 May 2011). Written informed consent was signed by parents (or legal guardian) and oral consent was obtained from children.

### 2.2. Dietary Data

Children’s diet (exposure) at the age of 4 years (evaluation wave of 2009–2010) was assessed by FFQ and applied to the child’s main caregiver by trained staff, querying about consumption frequency of 35 items during the previous 6 months. A detailed description on conversion into daily frequencies is described elsewhere [19]. Briefly, the frequencies present in the FFQ ranged from ≥4 times a day to never, and were converted to daily frequency of consumption (e.g., the mid-point of 1–3 times per month, twice a month, was divided by 30 days, and thus converted into 0.07 times per day). As described earlier [16,20], sixteen food items or groups were considered to identify *a posteriori* dietary patterns at 4 years of age: milk, yoghurt, cheese, meat/eggs, processed meat, fish, rice/pasta/potatoes, bread, fruit, vegetables, vegetable soup, sweets, sugar-sweetened beverages, crisps, fast food, and savory pastry. Considering little variability of consumption, food items/groups were categorized into first quintile, second to fourth quintiles aggregated, and fifth quintile. Regarding foods eaten only once or twice a day (e.g., vegetable soup), two categories were defined (1/day and 2/day). These cut-offs were used for the whole sample because cut-points were very similar between girls and boys at the age of 4 years.

For a subsample of 2373 participants, complete food diaries (two weekdays and one weekend day) were available, enabling evaluation of the FFQ validity and reliability. Considering Pearson’s correlation coefficients and intraclass correlation coefficients (ICC), comparison of dietary data assessed by FFQ or by food diary revealed weak-to-moderate correlations and fair-to-moderate agreement for foods with high frequency of consumption [20]. Moderate Pearson’s correlations were observed for vegetable soup (r = 0.54, *p* < 0.001), yoghurts (r = 0.48, *p* < 0.001), milk (r = 0.46, *p* < 0.001), and fruit (r = 0.42, *p* < 0.001). ICC ranged from 0.54 [95% confidence interval (95% CI) 0.51–0.56] for vegetable soup to 0.17 (95% CI 0.11–0.32) for juices. In addition, as described elsewhere [21], a z-score method was used to estimate the mean daily consumption as measured by the frequency reported in the FFQ in grams per day, calibrated by information obtained from the 3-day food diaries, showing that the FFQ is a reasonable measurement of dietary intake.

Since response options in the FFQ are categorical and due to their non-symmetrical distribution, children’s dietary patterns were identified by latent class analysis (LCA) [16,20]. This person-centered method distinguishes clusters of individuals from a sample [22]. Individuals were assigned to each dietary pattern in accordance with higher likelihood of membership to each identified class. Probabilities of choosing each item response (e.g., consumption categories), conditional on class membership, can be explained according to item profiles in each latent class (cluster).

As previously described [16,20], LCA identified three dietary patterns at 4 years of age (exposure): (i) a pattern with higher consumption of energy-dense foods (sweets, sugar-sweetened beverages, savory pastry, and processed meat) named ‘energy-dense foods’ (EDF); (ii) a pattern named ‘snacking’ that was lower in foods usually consumed at lunch and dinner (e.g., vegetables on a plate, fish, meat, rice/pasta/potatoes), but higher in intermediate foods typically eaten at snacking occasions; and (iii) a ‘healthier’ dietary pattern (used as reference category) followed by children with higher consumption of fruit, vegetables, vegetable soup, and fish, with a lower consumption of EDF.

### 2.3. Anthropometrics and Body Composition

Adiposity indicators considered at age of 10 years were the following; BMI standard deviation (SD) scores (BMI Z-scores), fat mass percentage (FM%), fat mass index (FMI), and waist-to-height ratio (WHtR).

Anthropometrics were evaluated by trained interviewers as previously described [12,17,20]. Briefly, children’s weight was measured to the nearest 0.1 kg (in light clothing) with a digital scale (Tanita^®^, Arlington Heights, IL, USA), and height was measured using a wall stadiometer (SECA^®^, Hamburg, Germany) to the nearest 0.1 cm.

At the ages of 4, 7, and 10 years, BMI z-scores were computed according to the WHO’s criteria [23,24]. At 4 and 7 years of age, BMI z-scores were used as continuous variables. BMI z-scores (outcome) when children were aged 10 years were considered for analyses, both as continuous and as categorical, the latter according to the WHO cut-points [24] (overweight, BMI z-score > 1 SD; obesity, BMI z-score > 2 SD; vs. normal, BMI z-score ≤ 1 SD). Children’s waist circumference was measured according to standard procedures [25], and WHtR (waist in centimeters / height in centimeters) at 10 years was further converted into z-scores with the sex-specific means and SD of the present sample, being considered as continuous in all analyses.

Body composition was evaluated by tetrapolar bioelectric impedance (BIA 101 Anniversary^®^, Akern, Florence, Italy) according to standard protocols. At 10 years of age, outputs were converted into fat-free mass using the equation proposed by Clasey [26], and fat mass percentage (FM%), as well as fat mass index [kg/m^2^ (FMI)], were computed in accordance. These variables were also converted into z-score units [mean = 0, SD = 1] using the sample’s sex-specific means and SD.

### 2.4. Cardiometabolic Risk Factors

At the 10-year evaluation wave, trained nurses collected venous blood samples in our research center after overnight fast and application of topical analgesic (EMLA cream). Samples were centrifuged (3500 rpm, 10 min) and were stored at −80 °C [12].

As detailed elsewhere [12], glucose was measured by ultraviolet enzymatic assay (hexokinase method), and insulin levels were determined by electrochemiluminescence immunoassay. Enzymatic colorimetric assay was used to measure TG and HDL-c, and the Friedewald equation [27] was used to compute LDL-c. HOMA-IR was calculated as ‘glucose (mg/dL) × insulin (µIU/mL)/405’. SBP and DBP were assessed with automatic sphygmomanometer (Model ELITE^®^, S Polo de Torrile, Italy) at the right brachial artery (two times, 5 min apart). For children with differences greater than 5 mmHg, a third assessment was made, and the mean of the two closest measurements was computed. Similar to adiposity measurements, cardiometabolic risk factors (outcomes) were also converted into z-scores [mean = 0, SD = 1], with sex-specific means and SD of the present sample, and were considered as continuous variables.

### 2.5. Potential Confounding Factors

Confounding factors evaluated were: maternal characteristics at child’s age of 4 years—education (number of completed schooling years), BMI, age (years), smoking (smoking 1–10 or >10 cigarettes/day vs. non-smoking), working status (working vs. not)— and child’s characteristics—sex, duration of non-excusive breastfeeding (<6 vs. ≥6 months), and exact age at the 4-year follow-up, as well as structured physical exercise practice (hours/week) and daily screen time (hours per day) as proxies of physical activity at 4 years of age]. In addition, at the 10-year wave, participant’s pubertal stage was evaluated by trained nurses using Tanner’s scale. In girls, breast development and pubic hair were assessed, whereas in boys, testicular development and pubic hair were assessed. Results from these evaluations were converted into stages (1 prepubertal, 2–4 pubertal, and 5 post-pubertal) [28,29].

### 2.6. Statistical Analysis

Associations of dietary patterns, followed at child’s 4 years, with the outcomes (BMI z-scores, FM%, FMI, W/Ht ratio, and other cardiometabolic risk factors at 10 years), were estimated by regression coefficients (β) and their respective 95% confidence intervals (95% CI) using linear regression models. In addition, multinomial regression models were used to estimate odds ratios (OR) and respective 95% CI for the association between dietary patterns and later overweight or obesity (defining children with normal BMI as reference).

Potential confounders were assessed one-by-one in each model, and those that did not change the associations of interest were not included in final models (maternal age, smoking status, and working status). Hence, results were adjusted for maternal education, maternal BMI, and child’s characteristics (BMI z-score at 4 years; exact age, non-exclusive breastfeeding duration; structured physical exercise, and daily screen time). These models were further adjusted for children’s BMI z-scores at the age of 7 years to assess if any hypothesized association was explained, totally or partially, by adiposity at this age.

The associations between dietary patterns (healthier pattern as reference) and adiposity (BMI, FM%, FMI, and WHtR) or other cardiometabolic risk factors (SBP, DBP, TG, HDL-c, LDL-c, glucose, insulin, and HOMA-IR) were examined in separate models for each of these outcomes.

Effect modification by child’s sex on the association of diet with adiposity, obesity, and cardiometabolic risk factors was examined by product terms of dietary pattern and child’s sex assessed in fully adjusted models.

Statistical analysis was conducted using SPSS statistical software package version 25 (SPSS Inc., Chicago, IL, USA), and a 5% significance level was adopted. Confidence intervals were computed taking into consideration Bonferroni’s correction for multiple post hoc tests.

## 3. Results

Mothers’ and children’s characteristics are shown in Table 1. The proportion of children following the EDF pattern at preschool age (4 years) was similar between girls and boys (41.4% and 41.6%, respectively), as was the mean BMI z-score (girls vs. boys: mean = 0.7; SD = 1.19; mean = 0.8; SD = 1.23) and WHtR (girls vs. boys: mean = 0.5; SD = 0.06; mean = 0.5; SD = 0.06) at 10 years of age. When compared to boys, girls showed a higher FM%, FMI, TG, LDL-c, fasting insulin, and HOMA-IR.

After adjustment for mothers’ and children’s characteristics (Table 2, Model 2), following an EDF dietary pattern at 4 years of age, when compared to following a healthier pattern (reference), was significantly positively associated with BMI (β = 0.139; 95% CI: 0.031, 0.246), FM% (β = 0.099; 95% CI: 0.002, 0.196), FMI (β = 0.129; 95% CI: 0.037, 0.221), and WHtR (β = 0.155; 95% CI: 0.055, 0.255) six years later only in girls (P-Interaction: 0.042; 0.113; 0.048; and 0.033, respectively). In models with a further adjustment for BMI z-scores at the age of 7 years (Model 3), the EDF pattern at 4 years of age remained significantly positively associated with most adiposity measures in girls.

When considering obesity (Table 3, Model 2), girls practicing the EDF dietary pattern at 4 years were significantly more likely to have obesity six years later (EDF vs. healthier: OR = 2.68; 95% CI: 1.55, 4.63). Again, a statistically significant effect modification according to sex was found on the association between diet and obesity (P-interaction = 0.019). These associations remained statistically significant after adjustment for BMI z-scores at 7 years of age (Model 3).

We also found a statistically significant effect modification by child’s sex on the association of the dietary pattern with glucose (P-interaction = 0.001), insulin (P-interaction = 0.033), and HOMA-IR (P-interaction = 0.018). Girls aged 4 years practicing the EDF dietary pattern, when compared to those following the healthier one, showed significantly higher insulin (β = 0.165; 95% CI: 0.020, 0.311) and HOMA-IR (β = 0.159; 95% CI: 0.013, 0.306) when they were 10 years old (Table 4, Model 2). No significant associations between the snacking dietary pattern and adiposity outcomes, or other cardiometabolic risk factors, were observed.

## 4. Discussion

This study shows that an EDF dietary pattern followed during preschool age is associated with increased adiposity, obesity, systolic blood pressure, and insulin resistance six years later in girls, highlighting sex-heterogeneity on the association of an early diet with both adiposity and cardiometabolic risk factors. It also shows that the BMI at 7 years of age only explains part of these associations, as, after its inclusion in the models, the magnitude of the associations of the dietary pattern at 4 years with adiposity, obesity, and cardiometabolic risk factors at 10 years remained significant and were slightly decreased in magnitude.

A dietary pattern comparable to the present EDF pattern was identified by reduced rank regression in the UK Avon Longitudinal Study of Parents and Children (ALSPAC), named ‘energy-dense, high-fat, low-fiber’. In a sample of 483 children from the ALSPAC cohort, this dietary pattern when followed at the age of 5 years was not associated with fat mass 4 years later, but, from 7 to 9 years of age, there was a positive significant association with fat mass [30]. These findings are only partially comparable to those of the current study for various methodological reasons, as well as because formal testing for sex-heterogeneity on the association of dietary pattern and adiposity was not mentioned, although the small sample size may have precluded such an analysis. In a larger sample of the ALSPAC cohort (*n* = 6772 children), the ‘energy-dense, high-fat, low-fiber’ dietary pattern followed at 7 years was significantly positively associated with FMI at 11 years [31], and effect modification was formally tested, but there was no evidence of sex-heterogeneity on the association between diet and adiposity.

In contrast, in a sample of 3911 children enrolled from the ALSPAC cohort, sex-heterogeneity was observed with a ‘health aware’ pattern (identified by principal component analysis) that was comparable to the present healthier pattern, that was negatively associated with fat mass gain in girls [32]. In addition, in an Iranian cross-sectional study that included 637 school-aged children, sex-heterogeneity was found on the association between dietary patterns and BMI, and girls with overweight were more likely to follow ‘Western’ or ‘sweet dairy’ dietary patterns [33].

Relative to cardiometabolic risk factors, few studies considering exposure as *a posteriori* dietary patterns were conducted in children and adolescents, most have a cross-sectional design, and those evaluating dietary patterns during preschool age are scarce [8,34]. In a cross-sectional study conducted in Australian adolescents [31], girls following a ‘Western’ dietary pattern showed an increased metabolic risk, whereas no association was found for boys. In Korean pre-pubertal children (8–9 years of age) a ‘Western’ pattern showed a positive association with a metabolic risk score in girls, but not in boys [35].

In the current cohort, we have previously observed sex-heterogeneity on the association between dietary patterns and adiposity, from the ages of 4 to 7 years, with the EDF pattern positively associated with adiposity (BMI, FMI, and WHtR) only in girls [16]. However, a recently published study with children from the same cohort did not find evidence of sex differences on the association between dietary pattern and cardiometabolic health from 7 to 10 years [12]. Yet, from 4 to 10 years, we have observed sex differences that are consistent for both adiposity and some cardiometabolic risk factors (insulin and insulin resistance). Different findings may be related to distinct methods for the identification of dietary patterns or to distinct age periods evaluated. In the ALSPAC cohort, *a posteriori* patterns, identified by distinct methods at different age periods, also found sex differences on the association with adiposity [30,31,32].

Finding that the EDF dietary pattern was associated with higher adiposity, insulin, and insulin resistance only in girls deserves further discussion. The current study confirms a positive association between an EDF dietary pattern followed during preschool age and later adiposity among girls, showing a persistence of this association into later ages and suggesting early consequences for cardiometabolic risk factors. As emphasized earlier [16], the association of the EDF pattern with higher adiposity found in girls may be explained by the sexual dimorphism in adiposity observed before puberty, which shows higher body fat and abdominal subcutaneous adiposity among girls. We tested an adjustment for the Tanner stage to assess if an earlier puberty onset in girls could explain the findings of the current study, but this adjustment did not change the associations of interest.

Associations of dietary patterns with BMI and WHtR show differences according to the child’s sex. When body composition measures were considered, the same sex-heterogeneity on the association with adiposity was found in the current sample. Moreover, we also evaluated the relationship of dietary patterns with free-fat mass, and the EDF pattern was significantly positively associated with the free-fat mass index only in girls (EDF vs. healthier: β = 0.139; 95% CI: 0.061, 0.217), although effect modification according to the child’s sex did not reach statistical significance (P-interaction = 0.118).

As highlighted in a review of the mechanistic pathways of sex differences in cardiovascular disease [14], sex differences have consistently been observed in both experimental and clinical studies, as well as in different conditions and distinct species. In animals, sex differences in response to exercise have been reported, with female mice showing increased free fatty acid levels and a reduced glucose uptake, as well as changes in cardiac substrate availability and utilization shifting to a greater use of fatty acids. In humans, in disease conditions such as heart failure, females usually have a higher expression of genes related to energy metabolism. Sex differences in gene expression are apparent at early ages, as shown in human umbilical vein endothelial cells. The sex-specific regulation of adipose triacylglycerols lipase further contributes to sex differences in energy metabolism, with increased rates of myocardial fatty acid oxidation among females.

In the current sample, as published earlier [17], results from likelihood ratio tests indicated that the effect of energy intake was independent of macronutrients in girls, but not for boys, among whom, protein and carbohydrates had independent effects on adiposity. Sex-specific associations could be explained by sex differences in the importance of fat versus protein and carbohydrates in metabolism. It has been proposed [36] that differences in sensitivity to central insulin and leptin between females and males, apparent since early ages, may reflect females’ physiological propensity to use fat as a substrate, whereas males rely more on glucose and protein metabolism.

Adiposity might be an intermediate step in the pathway between dietary intake and cardiometabolic risk factors. Hence, we adjusted models for BMI z-scores at 7 years of age to assess this possibility. However, although some associations were partially explained by adiposity at 7 years, among girls, direct effects of the EDF dietary pattern were still observed on adiposity and some cardiometabolic risk factors from 4 to 10 years of age.

If the sex-heterogeneity shown in the present study is true, current results may reflect a higher predisposition to store body fat and a higher risk of obesity reported in females [15,36,37]. However, given that males and females are both susceptible to obesity, and that girls may have an earlier adiposity rebound, it is possible that the current EDF pattern may show a relationship with adiposity in boys at later stages.

The assessment of dietary intake by FFQ could be regarded as a limitation. Yet, calibration with 3-day food diaries supports a reasonable validity of this methodology in assessing children’s dietary intake [38]. In addition, the EDF pattern was significantly higher in energy than the snacking and healthier patterns (girls, 6765 vs. 6304 and 6576 kJ; boys, 7175 vs. 6735 and 6910 kJ).

The inclusion of a subsample could implicate some bias into the analysis. However, a comparison of the present sample with the remaining cohort showed acceptable Cohen’s effect size values (<0.35). Hence, differences between this sample and the remaining cohort are probably related to a large sample size and not to relevant differences [38].

Assessment of several adiposity measures is an advantage of the present study, and using an index adjusted for height (i.e., FMI) is important, as the BMI is not able to distinguish between fat and lean mass, whereas the accuracy of FM% depends on both the height and free-fat mass [39]. In addition, the evaluation of several cardiometabolic risk factors enabled a more comprehensive analysis, which is relevant as children with obesity show an increased cardiovascular risk in adult life, exhibiting signs of cardiovascular dysfunction at early ages independent of other comorbidities, such as dyslipidemia or insulin resistance [40]. However, we were not able to evaluate cardiovascular parameters, such as left ventricular mass or intima-media thickness, that have been reported to be impaired in young children with overweight or obesity [41]. The prospective nature of this study, the age frame evaluated, and the assessment of several adiposity measures, plus different cardiometabolic risk factors, are considerable major strengths of this study. Furthermore, the sample size constitutes an important advantage that warrants assessment of effect modification by child’s sex. Moreover, as participants are part of a large cohort that is regularly followed, the assessment of several confounders was possible.

## 5. Conclusions

Current results suggest sex-heterogeneity on the association of dietary patterns with both adiposity and other cardiometabolic risk factors. A dietary pattern high in energy-dense foods practiced at preschool age is positively associated with later adiposity, overweight, and obesity in girls, in whom, it increases cardiometabolic risk factors 6 years later. In addition, there seems to be a direct effect of this early dietary pattern on adiposity and cardiometabolic risk factors at 10 years of age, independent of the BMI at 7 years, in girls. In contrast, among boys, an energy-dense dietary pattern at 4 years of age did not show an association with adiposity, nor with cardiovascular risk factors 6 years later.

These findings suggest that preventive strategies should be directed at preschool ages, with a special focus on energy-dense dietary patterns among girls. Future studies should further examine sex-heterogeneity on the association of diet with adiposity and cardiometabolic health in distinct populations and during different age periods.

## Figures and Tables

**Table 1 nutrients-14-00540-t001:** Demographic, dietary, anthropometric characteristics, and cardiometabolic markers by child’s sex.

	Characteristic	Girls, *n* = 1861	Boys, *n* = 1962
Maternal characteristics at 4 years, mean (SD)	Age (years)	34.5 (5.23)	34.4 (5.00)
	Education (years)	11.5 (4.25)	11.5 (4.13)
	BMI (kg/m^2^)	26.4 (5.22)	26.3 (4.92)
Child’s characteristics at 4 years, *n* (%)	Dietary pattern		
	EDF	771 (41.4)	816 (41.6)
	Snacking	242 (13.0)	300 (15.3)
	Healthier	848 (45.6)	846 (43.1)
Child’s adiposity at 10 years, mean (SD)	BMI z-score	0.7 (1.19)	0.8 (1.23)
	Fat mass (%)	23.2 (9.70)	21.2 (9.47)
	Fat mass index (kg/m^2^)	4.7 (2.80)	4.2 (2.63)
	Waist-to-height ratio	0.5 (0.06)	0.5 (0.06)
Child’s cardiometabolic markers at 10 years, mean (SD)	Systolic blood pressure (mmHg)	109.8 (9.76)	109.6 (9.34)
	Diastolic blood pressure (mmHg)	68.9 (7.02)	69.6 (7.22)
	Triacylglycerols (mg/dL)	71.9 (40.59)	62.2 (28.20)
	HDL-cholesterol (mg/dL)	54.4 (10.67)	56.7 (10.45)
	LDL-cholesterol (mg/dL)	95.3 (23.71)	92.7 (22.53)
	Glucose (mg/dL)	86.3 (6.18)	87.7 (9.11)
	Fasting serum insulin (µIU/mL)	10.8 (6.81)	8.0 (5.16)
	HOMA-IR	2.3 (1.56)	1.8 (1.21)

Footer: BMI, body mass index; EDF, energy-dense foods dietary pattern; BMI z-score, body mass index standard deviation scores according to the World Health Organization; HDL-c, high-density lipoprotein cholesterol; LDL-c, low-density lipoprotein cholesterol; HOMA-IR, homeostatic model assessment of insulin resistance.

**Table 2 nutrients-14-00540-t002:** Association of dietary patterns at 4 years of age with adiposity at 10 years of age, by child’s sex.

Adiposity at 10 Years	Dietary Pattern at 4 Years	Girls			Boys		
		n	ẞ	95% CI ^1^	n	ẞ	95% CI ^1^
BMI ^2^							
Model 1	Healthier	848	Ref		846	Ref	
	EDF	771	**0.228**	**0.077; 0.380**	816	0.031	−0.124; 0.186
	Snacking	242	−0.032	−0.254; 0.190	300	−0.038	−0.251; 0.174
Model 2	Healthier	848	Ref		846	Ref	
	EDF	771	**0.139**	**0.031; 0.246**	816	0.015	−0.103; 0.133
	Snacking	242	0.000	−0.153; 0.154	300	−0.015	−0.174; 0.145
Model 3	Healthier	848	Ref		846	Ref	
	EDF	771	**0.074**	**0.002; 0.146**	816	0.021	−0.055; 0.097
	Snacking	242	0.011	−0.091; 0.114	300	−0.001	−0.104; 0.103
FM% ^3^							
Model 1	Healthier	846	Ref		844	Ref	
	EDF	770	**0.186**	**0.058; 0.314**	809	0.039	−0.087; 0.165
	Snacking	240	0.026	−0.162; 0.214	297	−0.025	−0.198; 0.147
Model 2	Healthier	846	Ref		844	Ref	
	EDF	770	**0.099**	**0.002; 0.196**	809	0.015	−0.088; 0.117
	Snacking	240	0.041	−0.098; 0.179	297	−0.018	−0.157; 0.121
Model 3	Healthier	846	Ref		844	Ref	
	EDF	770	0.045	−0.026; 0.116	809	0.020	−0.053; 0.093
	Snacking	240	0.050	−0.051; 0.151	297	−0.004	−0.004; 0.095
FMI ^3^							
Model 1	Healthier	846	Ref		844	Ref	
	EDF	770	**0.217**	**0.089; 0.344**	809	0.051	−0.075; 0.177
	Snacking	240	0.053	−0.134; 0.241	297	−0.028	−0.200; 0.144
Model 2	Healthier	846	Ref		844	Ref	
	EDF	770	**0.129**	**0.037; 0.221**	809	0.023	−0.077; 0.123
	Snacking	240	0.071	−0.060; 0.203	297	−0.021	−0.157; 0.115
Model 3	Healthier	846	Ref		844	Ref	
	EDF	770	**0.078**	**0.011; 0.145**	809	0.029	−0.040; 0.098
	Snacking	240	0.080	−0.016; 0.176	297	−0.007	−0.101; 0.087
WHtR ^3^							
Model 1	Healthier	847	Ref		845	Ref	
	EDF	771	**0.244**	**0.117; 0.372**	816	0.062	−0.064; 0.188
	Snacking	241	0.041	−0.147; 0.228	300	0.076	−0.097; 0.248
Model 2	Healthier	847	Ref		845	Ref	
	EDF	771	**0.155**	**0.055; 0.255**	816	0.021	−0.082; 0.124
	Snacking	241	0.041	−0.101; 0.184	300	0.059	−0.081; 0.199
Model 3	Healthier	847	Ref		845	Ref	
	EDF	771	**0.108**	**0.028; 0.187**	816	0.025	−0.050; 0.100
	Snacking	241	0.051	−0.061; 0.164	300	0.069	−0.033; 0.171

Footer: ẞ, linear regression coefficient; 95% CI, 95% confidence interval; EDF, energy-dense foods dietary pattern; Ref, reference category; BMI, body mass index standard deviation score; FM%, fat mass percent; FMI, fat mass index; WHtR, waist-to-height ratio. ^1^ Considering Bonferroni’s correction for multiple post hoc tests. ^2^ Age- and sex-specific BMI z-scores were defined according to the World Health Organization. ^3^ Adiposity measure converted into z-scores using the sex-specific mean and standard deviation of the current sample. Model 1 is crude. Model 2 is adjusted for maternal (education and body mass index at 4 years of age) and child’s (breastfeeding, weekly time practicing structured physical, screen time, exact age, and BMI z-scores at 4 years of age) characteristics. Model 3 includes variables in model 2 and is further adjusted for BMI z-scores at 7 years of age.

**Table 3 nutrients-14-00540-t003:** Association of dietary patterns at 4 years of age with overweight or obesity at 10 years of age, by child’s sex.

		Girls (*n* = 1861)						Boys (*n* = 1962)					
		Overweight (*n* = 503) ^1,2^			Obesity (*n* = 272) ^1,2^			Overweight (*n* = 490) ^1,2^			Obesity (*n* = 357) ^1,2^		
		*n*	OR	95% CI ^3^	*N*	OR	95% CI ^3^	*n*	OR	95% CI ^3^	*n*	OR	95% CI ^3^
Model 1 ^4^	Healthier	229	Ref		88	Ref		218	Ref		143	Ref	
	EDF	216	1.24	0.92; 1.67	150	**2.24**	**1.52; 3.29**	196	0.95	0.70; 1.29	162	1.20	0.85; 1.69
	Snacking	58	0.90	0.57; 1.40	34	1.37	0.77; 2.42	76	0.98	0.65; 1.49	52	1.03	0.64; 1.65
Model 2 ^5^	Healthier	229	Ref		88	Ref		218	Ref		143	Ref	
	EDF	216	1.32	0.92; 1.88	150	**2.68**	**1.55; 4.63**	196	0.98	0.69; 1.37	162	1.13	0.73; 1.75
	Snacking	58	1.07	0.63; 1.80	34	1.65	0.72; 3.74	76	1.02	0.65; 1.61	52	1.01	0.55; 1.86
Model 3 ^6^	Healthier	229	Ref		88	Ref		218	Ref		143	Ref	
	EDF	216	1.48	0.96; 2.29	150	**3.15**	**1.54; 6.45**	196	1.01	0.66; 1.53	162	1.16	0.62; 2.18
	Snacking	58	1.32	0.67; 2.58	34	1.91	0.65; 5.60	76	1.04	0.58; 1.86	52	0.86	0.36; 2.07

Footer: EDF, energy-dense foods dietary pattern; Ref, reference category; OR, odds ratio; 95% CI, 95% confidence interval. ^1^ Age- and sex-specific BMI z-scores were categorized into overweight (>1 SD) or obesity (>2 SD) according to the World Health Organization. ^2^ Children not overweight nor obese (girls, *n* = 1076; boys, *n* = 1100) were considered as refence category. ^3^ Considering Bonferroni’s correction for multiple post hoc tests. ^4^ Model 1 is crude. ^5^ Model 2 is adjusted for maternal (education and body mass index at 4 years of age) and child’s (breastfeeding, weekly time practicing structured physical, screen time, exact age, and BMI z-scores at 4 years of age) characteristics. ^6^ Model 3 includes variables in model 2 and is further adjusted for BMI z-scores at 7 years of age.

**Table 4 nutrients-14-00540-t004:** Association of dietary patterns at 4 years of age with cardiometabolic markers at 10 years of age, by child’s sex ^1^.

Cardiometabolic Risk Factors at 10 Years	Dietary Pattern at 4 Years	Girls			Boys		
		*N*	ẞ	95% CI ^2^	*n*	ẞ	95% CI ^2^
SBP							
Model 1 ^3^	Healthier	847	Ref		844	Ref	
	EDF	771	**0.179**	**0.055; 0.303**	816	0.056	−0.068; 0.180
	Snacking	242	0.114	−0.069; 0.296	300	−0.060	−0.229; 0.108
Model 2 ^4^	Healthier	847	Ref		844	Ref	
	EDF	771	0.123	−0.004; 0.249	816	0.029	−0.095; 0.153
	Snacking	242	0.092	−0.089; 0.272	300	−0.082	−0.249; 0.085
Model 3 ^5^	Healthier	847	Ref		844	Ref	
	EDF	771	0.108	−0.019; 0.236	816	0.034	−0.087; 0.156
	Snacking	242	0.090	−0.091; 0.271	300	−0.081	−0.241; 0.078
DBP							
Model 1 ^3^	Healthier	844	Ref		837	Ref	
	EDF	769	**0.126**	**0.002; 0.251**	808	0.056	−0.069; 0.180
	Snacking	239	0.145	−0.038; 0.327	296	−0.037	−0.206; 0.133
Model 2 ^4^	Healthier	844	Ref		837	Ref	
	EDF	769	0.062	−0.065; 0.189	808	0.006	−0.120; 0.132
	Snacking	239	0.103	−0.078; 0.285	296	−0.090	−0.260; 0.080
Model 3 ^5^	Healthier	844	Ref		837	Ref	
	EDF	769	0.039	−0.090; 0.168	808	0.005	−0.120; 0.130
	Snacking	239	0.111	−0.072; 0.294	296	−0.080	−0.250; 0.090
TAG							
Model 1 ^3^	Healthier	588	Ref		625	Ref	
	EDF	569	0.137	−0.012; 0.287	618	0.058	−0.086; 0.202
	Snacking	168	0.074	−0.148; 0.296	220	−0.026	−0.223; 0.171
Model 2 ^4^	Healthier	588	Ref		625	Ref	
	EDF	569	0.106	−0.048; 0.261	618	0.047	−0.101; 0.195
	Snacking	168	0.079	−0.146; 0.304	220	−0.032	−0.232; 0.169
Model 3 ^5^	Healthier	588	Ref		625	Ref	
	EDF	569	0.075	−0.081; 0.231	618	0.047	−0.102; 0.196
	Snacking	168	0.103	−0.124; 0.330	220	−0.032	−0.236; 0.172
HDL-c							
Model 1 ^3^	Healthier	588	Ref		625	Ref	
	EDF	569	−0.061	−0.208; 0.087	618	−0.111	−0.256; 0.033
	Snacking	168	0.110	−0.109; 0.330	220	0.035	−0.163; 0.232
Model 2 ^4^	Healthier	588	Ref		625	Ref	
	EDF	569	−0.040	−0.192; 0.112	618	−0.072	−0.220; 0.075
	Snacking	168	0.094	−0.127; 0.315	220	0.071	−0.128; 0.270
Model 3 ^5^	Healthier	588	Ref		625	Ref	
	EDF	569	−0.017	−0.171; 0.137	618	−0.073	−0.223; 0.076
	Snacking	168	0.075	−0.149; 0.299	220	0.056	−0.149; 0.260
LDL-c							
Model 1 ^3^	Healthier	588	Ref		625	Ref	
	EDF	569	−0.011	−0.158; 0.136	618	−0.028	−0.172; 0.116
	Snacking	168	−0.183	−0.402; 0.036	220	−0.055	−0.252; 0.142
Model 2 ^4^	Healthier	588	Ref		625	Ref	
	EDF	569	0.014	−0.140; 0.167	618	−0.018	−0.167; 0.131
	Snacking	168	−0.133	−0.356; 0.091	220	−0.044	−0.245; 0.158
Model 3 ^5^	Healthier	588	Ref		625	Ref	
	EDF	569	0.004	−0.155; 0.163	618	−0.012	−0.162; 0.138
	Snacking	168	−0.131	−0.362; 0.100	220	−0.034	−0.239; 0.170
Glucose							
Model 1 ^3^	Healthier	582	Ref		616	Ref	
	EDF	565	**0.162**	**0.016; 0.308**	611	−0.118	−0.272; 0.036
	Snacking	167	−0.037	−0.253; 0.180	218	−0.062	−0.274; 0.150
Model 2 ^4^	Healthier	582	Ref		616	Ref	
	EDF	565	0.099	−0.053; 0.251	611	−0.095	−0.255; 0.065
	Snacking	167	−0.110	−0.332; 0.111	218	−0.034	−0.251; 0.183
Model 3 ^5^	Healthier	582	Ref		616	Ref	
	EDF	565	0.099	−0.058; 0.256	611	−0.110	−0.274; 0.054
	Snacking	167	−0.102	−0.331; 0.126	218	−0.052	−0.277; 0.173
Insulin							
Model 1 ^3^	Healthier	582	Ref		616	Ref	
	EDF	565	**0.239**	**0.093; 0.386**	611	0.024	−0.119; 0.167
	Snacking	167	0.088	−0.130; 0.305	218	0.084	−0.112; 0.281
Model 2 ^4^	Healthier	582	Ref		616	Ref	
	EDF	565	**0.165**	**0.020; 0.311**	611	0.007	−0.136; 0.150
	Snacking	167	0.065	−0.147; 0.277	218	0.082	−0.112; 0.277
Model 3 ^5^	Healthier	582	Ref		616	Ref	
	EDF	565	0.140	−0.001; 0.282	611	0.013	−0.125; 0.152
	Snacking	167	0.111	−0.094; 0.316	218	0.102	−0.088; 0.292
HOMA-IR							
Model 1 ^3^	Healthier	582	Ref		616	Ref	
	EDF	565	**0.238**	**0.092; 0.384**	611	0.003	−0.140; 0.146
	Snacking	167	0.080	−0.138; 0.297	218	0.072	−0.125; 0.268
Model 2 ^4^	Healthier	582	Ref		616	Ref	
	EDF	565	**0.159**	**0.013; 0.306**	611	−0.010	−0.154; 0.134
	Snacking	167	0.048	−0.165; 0.261	218	0.074	−0.121; 0.269
Model 3 ^5^	Healthier	582	Ref		616	Ref	
	EDF	565	0.135	−0.008; 0.278	611	−0.006	−0.146; 0.133
	Snacking	167	0.092	−0.115; 0.300	218	0.090	−0.102; 0.282

Footer: ẞ, linear regression coefficient; 95% CI, 95% confidence interval; SBP, systolic blood pressure, DBP, diastolic blood pressure, TAG, triacylglycerols; HDL-c, high-density lipoprotein cholesterol; LDL-c, low-density lipoprotein cholesterol; HOMA-IR, homeostatic model assessment of insulin resistance; EDF, energy-dense foods dietary pattern; Ref, reference category. ^1^ All cardiometabolic markers were converted into z-scores using the sex-specific means and standard deviations of the sample. ^2^ Considering Bonferroni’s correction for multiple post hoc tests. ^3^ Model 1 is crude. ^4^ Model 2 is adjusted for maternal (education and body mass index at 4 years of age) and child’s (breastfeeding, weekly time practicing structured physical, screen time, exact age, and BMI z-scores at 4 years of age) characteristics. ^5^ Model 3 includes variables in model 2 and is further adjusted for BMI z-scores at 7 years of age.

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
