# Peer review of "Sex-Heterogeneity on the Association between Dietary Patterns at 4 Years of Age with Adiposity and Cardiometabolic Risk Factors at 10 Years of Age"

_nutrients, 2022, doi:10.3390/nu14030540_

Round 1

Reviewer 1 Report

  1. Description of the studied population is very inaccurate. Who were these children? Where were they examined? Where this study was conducted? More details are needed.
  2. The Authors wrote that "This prospective study was based on the population-based birth cohort Generation XXI 67 (Porto, Portugal, 2005-2006), described elsewhere [18]. But these are data from over 15 years ago. Later they wrote that "The current analysis considers singleton children" and that "Details on response options and con version into daily frequencies are described elsewhere [19]". But Reference position no. 19 reffers to a paper from 2015, so how can we say that these are current analysis? I supose that the study reffers to a population which was investigated several years ago, but it must be clearly stated. The Authors must clearly state when exacly the population was examined, what was done and already published and what is new? It seems to me that the present paper is a new data-analysis on the data obtained several years ago. Even if we decide that it is ok, and that the results are interesting and worth to be published, it must be clearly described and stated. In the present form I do not understand what is new and what is old.
  3. line 95 "In a subsample of 2373 children" - what is subsample? Is this the 10-year-old population of children on which the study was currently conducted?

        I suppose the study was on a population that is now 10 years old (and a former investigations were at 4 years of age). But it  is not clear to me if it was the exact same population of children. If so, how was this ensured with so many participants?

4. line 129 - the Authors should describe, even in a brief way, how the anthropometric parameters were evaluated. Total cross-referencing is troublesome for the readers.

5. line 395 - in the Conclusions the Authors wrote that "girls may need a particularly earlier attention". And what about boys? There is nothing about boys in the summary.

In general, the manuscript is interesting and maybe worth to be published. However, in the present shape, it does not have appropraite quality to be published. Especially, the materials and methods section must be improved, regarding composition of the studied population, and a clear identification what was already published and what is new. Details I have pointed above.

Author Response

We thank you very much for your comments. We attached a word file with a point-by-point response. Please see attachment.

Reviewer 2 Report

The paper entitled “Sex-heterogeneity on the association between dietary patterns at 4 years of age with adiposity and cardiometabolic risk factors at 10 years of age” is a well-designed longitudinal study, with a large sample size. There are only some points to be improved and clarified.

-       The authors talked about cardiometabolic factors, but actually, they did not evaluate cardiovascular parameters, such as left ventricular mass or intima-media thickness, that are reported to be impaired already in young children with overweight/obesity (Drozdz D et al. Obesity and Cardiometabolic Risk Factors: From Childhood to Adulthood. Nutrients. 2021 Nov 22;13(11):4176. Genoni G et al. Healthy Lifestyle Intervention and Weight Loss Improve Cardiovascular Dysfunction in Children with Obesity. Nutrients. 2021 Apr 15;13(4):1301. Cote AT et al. Childhood obesity and cardiovascular dysfunction. J Am Coll Cardiol. 2013 Oct 8;62(15):1309-19.). If not taken into account in the design phase of the study, it should at least be discussed in the discussion section of the paper.

-       Can you add data about children with celiac disease? Were there any? Were they excluded?

-       What about breakfast skipping and adiposity? This aspect should be considered and discussed.

-       In the discussion, the authors stated they performed an adjustment for Tanner stage, but no assessment of pubertal status is mentioned in the methods section.

Author Response

Thank you very much for your comments. Please see attachement with a point-by-point answer. 

Round 2

Reviewer 1 Report

The manuscript was significantly improved. The authors explained and corrected many parts of the manuscript which I had doubts about.

  1. line 99 - I do not understand why the Authors added the term "exposure". Exposure is connected with a "dosage" and very often with contact to raher negative factors (like radiation, toxic heavy metals, pesticides). Dietary intake of nutrients is not exposure. It is better to remove this term.
  2. line 425 - I still have doubts regarding conclusions formulated for boys. The fragment "However, in boys, similar associations may be observable at later ages, and future studies should further confirm this sex heterogeneity on the association of diet with adiposity and cardiometabolic health in distinct populations and during different age periods." indeed says are nothing and in the title we have "sex-heterogenity", so I advise to once again modfy this fragment and put information that "based on the obtained results similar associations cannot be observed for boys at the investigated age, but might be found at later ages"

Author Response

Thank you very much for your commnts. Please find our answers in the attachment. You hope we have adressed all your concerns. 
